# The Intersection of Serine Metabolism and Cellular Dysfunction in Retinal Degeneration

**DOI:** 10.3390/cells9030674

**Published:** 2020-03-10

**Authors:** Tirthankar Sinha, Larissa Ikelle, Muna I. Naash, Muayyad R. Al-Ubaidi

**Affiliations:** Department of Biomedical Engineering, University of Houston, Houston, TX 77204, USA; tsinha2@Central.UH.EDU (T.S.); likelle@Central.UH.EDU (L.I.)

**Keywords:** serine, retinal degeneration, diabetic retinopathy, macular degeneration, macular telangiectasia, oxidative stress, sphingolipids, retina, RPE, Müller cells

## Abstract

In the past, the importance of serine to pathologic or physiologic anomalies was inadequately addressed. Omics research has significantly advanced in the last two decades, and metabolomic data of various tissues has finally brought serine metabolism to the forefront of metabolic research, primarily for its varied role throughout the central nervous system. The retina is one of the most complex neuronal tissues with a multitude of functions. Although recent studies have highlighted the importance of free serine and its derivatives to retinal homeostasis, currently few reviews exist that comprehensively analyze the topic. Here, we address this gap by emphasizing how and why the de novo production and demand for serine is exceptionally elevated in the retina. Many basic physiological functions of the retina require serine. Serine-derived sphingolipids and phosphatidylserine for phagocytosis by the retinal pigment epithelium (RPE) and neuronal crosstalk of the inner retina via D-serine require proper serine metabolism. Moreover, serine is involved in sphingolipid–ceramide balance for both the outer retina and the RPE and the reductive currency generation for the RPE via serine biosynthesis. Finally and perhaps the most vital part of serine metabolism is free radical scavenging in the entire retina via serine-derived scavengers like glycine and GSH. It is hard to imagine that a single tissue could have such a broad and extensive dependency on serine homeostasis. Any dysregulation in serine mechanisms can result in a wide spectrum of retinopathies. Therefore, most critically, this review provides a strong argument for the exploration of serine-based clinical interventions for retinal pathologies.

## 1. Why Is Serine Important to the Entire Retina?

Serine is a non-essential amino acid directly involved in cellular homeostasis, proliferation, and differentiation [1,2]. The cells of the neural retina are no exception, and, in fact, exhibit a great dependence on serine and its exhaustive variety of metabolic intermediates [3]. Serine uptake occurs either by delivery from the bloodstream or it can be synthesized by the anabolism of the glycolytic intermediate, 3-phosphoglycerate (3-PG) [4] in the retina (neural retina-RPE, Figure 1). After uptake or synthesis, serine becomes available and serves as a central node in many cellular processes [5].

Other than being an integral amino acid in multiple essential proteins, free serine is essential for generating cysteine, glycine, methionine, and sphingolipids [5]. Glycine and cysteine are essential intermediates. Glycine is a precursor molecule for porphyrins and purine nucleotides [6]. Cysteine on the other hand is important in the protection of neuronal cells and for the production of taurine [7]. Together they form glutathione (GSH), a critical anti-oxidant in the retina [8]. Sphingolipids are elemental components of the phospholipid bilayer and are indispensable to cellular viability, homeostasis and function [2,3,5]. Furthermore, serine derived-metabolites have proven essential to methylation [2], apoptosis, and synaptic receptor activation [6]. 

The integrality of serine to cellular function has been appreciated by dysregulatory events appearing in many retinopathies. Reduced serine levels have been implicated in the etiology of macular related diseases such as macular telangiectasia type 2 (Mac Tel), age related macular degeneration, and diabetic retinopathy (DR) [9,10]. The retina is a complex stratified tissue consisting of the retinal pigment epithelium (RPE), a critical layer of cells for retinal homeostasis, and the neural retina containing the two types of photoreceptor cells. The neural retina also harbors the second order neurons and the retinal ganglion cells (RGC) that form the optic nerve. Serine proves to be a vital intermediate in many of these processes. Consequently, our goal is to provide a focused review of the role of serine homeostasis in maintaining optimum retinal function and proper oxidative balance.

## 2. Why Does the Retina Need to Synthesize Serine?

In most tissues, serine uptake from either blood or proteolysis sufficiently meets cellular metabolic requirements. However, there are tissues that mandate an elevated level of serine and, accordingly, upregulate enzymes for serine biosynthesis. As previously highlighted, this biosynthetic pathway (Figure 1) helps in maintaining the redox potential of the cell [11]. The primary source of serine biosynthesis is glucose [12], which in most cells is utilized via glycolysis. Serine biosynthesis branches from one of the glycolytic intermediates, 3-phosphoglycerate. The rate limiting step for serine biosynthesis is the reaction involving phosphoglycerate dehydrogenase (PHGDH), which converts 3-phosphoglycerate into phosphoserine. This is followed by the removal of the phosphate to generate l-serine.

It has been previously shown that the retina expresses high levels of all enzymes involved in serine biosynthesis. However, the RPE appears to express even higher levels than the neural retina [13] as has been shown by flux studies, whereby the RPE readily converts glucose into serine [10,14]. Further experiments verified that the neural retina possess an efficient system for serine uptake. [15].

### De Novo Serine Synthesis in Müller Cells

Transport of serine, a neutral amino acid, across the tight blood–retinal barrier into the RPE or across the endothelial cells and to the neural retina is supposedly inadequate [16]. So the retina increases the levels of intracellular serine through de novo synthesis [12]. Further supply of serine to photoreceptors and the inner retina is provided by the RPE and retinal Müller cells [15]. The latter was demonstrated by co-immunofluorescence with anti-PHGDH and anti-cellular retinaldehyde–binding protein (CRALBP) antibodies showing that the serine de novo synthesis pathway is indeed present in the Müller cells [17]. Since serine metabolism is central to maintaining redox and oxidative balance, ion flux, glutamate levels, and many other support functions [18], Müller cells through their de novo synthesis of serine play major roles in these functions [17].

Many retinopathies are associated with loss of Müller glia [17]. Mac Tel, is a pathology of the retina recently characterized by significant reductions in serine synthesis and loss of central vision [16]. Although the macula (anatomic) is a small cone-dense region of the retina occupying only 1.4% of the total area of the retina, it harbors approximately 8% of the total cone population and 60% of all RGCs [19]. The macula is incredibly metabolically active and relies heavily on Müller glia [18]. Not only has localization of serine synthesis in the neural retina been demonstrated in Müller cells, but relative to peripheral Müller cells, those of the macula seem to show increased expression of PHGDH [10]. Moreover, the macular Müller cells show increased GSH and glycine production and are more susceptible to induced stress [10].

## 3. Serine Homeostasis Plays an Important Role in the Maintenance of the Retina

### 3.1. Serine and Sphingolipids

One of the many fates of biosynthesized L-serine is to combine with palmitoyl-Co-A to form sphingolipids (Figure 2) catalyzed by the enzyme serine palmitoyl-Co-A transferase (SPT) [20]. Even though there are multiple interconnected pathways, which can control the formation of various sphingolipids, the most common pathway is serine incorporation [21]. Neural retinal sphingolipids have been well characterized addressing their beneficial and toxic capacities [3,22,23,24,25]. It is very well known that the most vital role of sphingolipids is aiding in sphingomyelin formation, which enables efficient synaptic transmission [3]. Perhaps, that is one reason why the sphingolipid content in the inner retina is quite high [3]. Moreover, sphingolipids in the form of sphingosine-1-phosphate are thought to have anti-apoptotic role, further rationalizing the abundance of sphingolipids in the outer retina [3]. It is important to note that the role of sphingolipid levels and their derivatives have not yet been assessed in the RPE. Given that the RPE may be a vital source of serine for the neural retina, it is imperative to determine how its transport might be facilitated.

### 3.2. Serine and RPE Phagocytosis

The efficient phagocytosis of photoreceptor outer segments by the RPE is elemental to retinal health. From extensive examination of the process, it has been determined that any delay or inefficiency in the phagocytosis can lead to gross abnormalities for both the neural retina and the RPE [26]. The metabolism of the RPE is largely dependent on recycling the outer segment phospholipids, as they provide an important source of fuel [27,28,29]. The neural retina eliminates older disks in order to maintain optimal function. So how is serine vital for maintaining such an important step? In order for the RPE to recognize and phagocytose the correct portion of the outer segments, phosphatidylserine will localize to the extracellular surfaces of those digestible regions [30]. Since almost 10% of the entire outer segment is daily phagocytosed from each photoreceptor, it helps us appreciate the enormous amount of serine that needs to be available. Furthermore, it has been suggested that sphingolipids may also play a regulatory role in order to ensure efficient phagocytosis, since disruption of sphingolipid metabolism via SPT and ceramide synthase inhibition impaired phagocytosis [31,32]. In addition, glycosphingolipids like lactosylceramides and gangliosides are major lipid raft components and assist in cell adhesion, membrane polarity and phagocytosis initiation [33,34,35,36]. This raises the possibility that the sphingolipid pool in RPE may assist in facilitating outer segment phagocytosis as well as maintaining the tight junction barrier and cellular polarity. Since serine is an integral component of sphingolipids, it goes without saying that this presents another facet of serine availability contributing to efficient neural retina:RPE interdependence.

### 3.3. The Role of D-Serine

D-serine, an enantiomer of L-serine, is primarily released by glial cells, Müller cells and astrocytes [13]. Initially, it was discovered in the brain but has more recently been studied in the neural retina [13]. D-serine, formed by the racemization of L-serine by serine racemase (SRR) [13], functions as a neurotrophic factor and as a co-agonist for *N*-methyl-D-aspartate receptors (NMDARs) [37]. In the brain, NMDARs normally bind glutamate and glycine, but data suggests preferential or higher affinity binding to D-serine in the retina [13,38]. NMDARs locate at the synaptic terminals of RGCs and sparingly to photoreceptor terminals [39]. The role of NMDARs in the synaptic terminals has not been entirely substantiated, however, studies have indicated that they function in an excitatory role [13]. In the presence of D-serine, NMDA-mediated currents and light-evoked response showed increased amplitude over the D-serine absent control [13]. Irrespective of NMDAR functions, the importance of D-serine is evidenced by pathologies that are directly linked to D-serine irregularities. D-serine insufficiency has been associated with psychiatric and neurodevelopmental disorders [37]. Supplementation with D-serine has proven to mitigate some of the symptoms of psychosis [37]. Patients with DR, by contrast, suffer from increased SRR activity and over production of D-serine, which elicits an excitotoxic effect on RGCs and ultimately contributes to cell death [40]. This will be addressed in more detail in the following sections.

### 3.4. Serine and Epigenetic Regulation

One carbon metabolism is a fundamental process for purine, thymidine, and amino acid biosynthesis that involves the transfer of a one carbon unit to generate critical metabolites [8]. Serine is essential in this group of metabolic reactions [41] and, in this context, is a precursor for the synthesis of methionine. Briefly, serine donates a methyl group which reacts with homocysteine, originally derived from aspartate [42,43]. This reaction is catalyzed by serine transhydroxymethylase to ultimately form methionine [42,43].

An adenosylation reaction of adenosine triphosphate (ATP) and methionine will yield *S*-adenosyl-L-methionine, which is the critical methyl donor for global methylation of DNA [41,44,45]. DNA methylation, an absolutely essential process to maintain cellular homeostasis [46], occurs on CpG islands of DNA and suppresses gene expression [47]. Therefore, irregularities in serine levels cause downstream hypo- or hyper-methylation of DNA [48], effecting stress responses, proliferation, metabolism, and even responses to extrinsic stimuli [46].

The link between epigenetic modulations and retinal disease is currently at a nascent phase [49], so correlations between aberrant serine metabolism, DNA methylation, and retinal disease remain unclear. However, recent studies performed on three pairs of monozygotic twins with different presentations of AMD indicated significant changes to DNA methylation patterns of genes that may be implicated in disease pathogenesis [50]. More critically, the diet of the studied twins also highlighted nutritional significance in epigenetic regulation. Subjects with reduced dietary methionine, vitamin D, and betaine had worse disease prognosis [50], implicating the importance of nutrient bioavailability and epigenetics. As serine is involved in the synthesis of methionine, further studies should explore serine levels in retinal disease in relation with epigenetic changes that may contribute to disease onset and/or progression.

## 4. Serine Is an Anti-Oxidant and Mediator of Inflammation

The role that serine plays as a central junction for critical intermediates also extends to anti-oxidative and inflammatory mechanisms. Experimental evidence has indicated that the retina shares some metabolic patterns with neoplastic cells [16]. In these systems, the Warburg effect predominates [51], and energy is generated primarily through glycolysis to ensure accelerated ATP production [16]. As explained earlier, the retina is one of the highest energy consuming tissues and thus a source of excessive reactive oxygen species (ROS). Emerging data has shed light on extra-mitochondrial sources of ROS specifically in the outer retina and even in the outer segment of photoreceptors [52,53]. It was previously shown that oxygen is mainly absorbed at the level of the photoreceptors [54] and thus this site is most prone to oxidative stress, both in physiology and pathology [55]. To build on that, it seems that blue light exposure on outer segment discs as well as ectopic oxidative phosphorylation in the outer segment both pose a greater need for effective free radical scavengers in the outer segment–RPE interface [56,57,58]. Glycine and GSH and thus serine supply are essential for this ROS mitigation system of the rods and cones [59].

Fundamentally, intracellular redox homeostasis represents the equilibrium between oxidative and anti-oxidative species, creating a balance in which the environment is not cytotoxic and is sensitive enough to redox changes that may mandate an intracellular or extracellular response. Clearly, oxidative stress is caused by aberrations to this balance. Serine metabolism contributes to redox homeostasis through synthesis of glycine and its essential downstream products GSH and nicotinamide adenine dinucleotide phosphate (NADPH), as well as nicotinamide adenine dinucleotide (NADH) generation during serine biosynthesis [60]. In addition, GSH and NADPH deactivate ROS and other oxidative molecules [60]. GSH is the direct result of combining cysteine and glycine, both of which, as previously explored, are synthesized from serine [61]. NADPH is generated in many pathways, but recent studies suggest that metabolism of serine is a significant contributor of NADPH to the mitochondria, especially during hypoxic conditions [11]. During serine synthesis from 3-phosphoglycerate, serine donates a single carbon to folate forming glycine as well as tetrahydrofolate (THF). THF reductase forms NADP^+^. Then, GSH mediated-reduction of electrophilic molecules helps to maintain the balance of NADPH to NADP^+^, which is principle to redox homeostasis.

GSH anti-oxidative activity is vital for proper retinal function [62]. So in malignancies where serine is deficient, there is an obvious reduction in GSH level and activity, and, as a consequence, the supply of the precursor molecules have been significantly affected. However, it is not as simple as reduced supply. 5’ adenosine monophosphate-activated protein kinase (AMP-K) can also influence the availability of GSH. AMP-K, like nuclear factor erythroid-derived 2-like 2 (NRF-2), is a “cellular sensor”, and responds to metabolic and redox irregularities [63]. AMP-K supports cell survival by upregulating anti-oxidant molecules such as GSH. Elevating serum serine levels in mice fed high fat diet showed increased levels of phosphorylated AMP-K, which resulted in reduced oxidative side-effects and increased GSH expression [63]. AMP-K and NRF-2 are, by no means, the only transcription factors involved in moderating the anti-oxidative response, but they represent the profound integration of serine metabolism and retinal oxidative homeostasis.

NRF-2 is an important redox-activated transcription factor under conditions of stress and imbalance of damaging oxidative species [64]. NRF-2 targets critical anti-oxidant genes such as superoxide dismutase, catalase, and GSH by upregulating transcription. In culture of non-small cell lung cancer cells (NSCLCs), gene enrichment analysis has demonstrated a correlation between PHGDH, the enzyme necessary for shunting 3-phosphoglycerate to serine synthesis, and genes that target the activation of NRF-2 [64]. The correlation suggests that biosynthesis of serine is involved in the expression and regulation of essential anti-oxidant proteins [64]. In addition to the cohort of anti-oxidant proteins, NRF-2 also has some regulatory involvement in the bioavailability of nitric oxide (NO) [65], which has the ability to mitigate the effects of H_2_O_2_ and superoxide [66]. Treatment with serine increases NO levels in culture which directly links NO production to serine [65], whether endogenously or exogenously presented.

Oxidative stress can also elicit tremendous damage to membranes, DNA, and mitochondria. However, in many pathologies, oxidative stress and inflammation are intimately interlinked [67]. In particular, AMD and DR have been characterized by the slow infiltration of pro-inflammatory constituents of the innate immune system [68]. Serine is currently being explored as a possible therapy for addressing inflammation in these pathologies. However, the involvement of serine metabolism in inflammation and innate immunity remains unclear [69].

Interesting observations have demonstrated a contradictory picture. Exogenously administered serine has been shown to reduce levels of inflammatory cytokines such as interleukin-1β and interleukin-6 [63]. Serine has also been shown to weaken the pro-inflammatory response necessary for macrophage recruitment after bacterial infections in mice [70]. The conflict emanates from the upregulation of inflammatory elements by way of increased glycolytic flux. In DR and AMD, dietary mismanagement and genetic mutations lead to metabolic dysregulation [71,72,73,74], which may increase glycolysis in cells. This may lead to increased synthesis of serine since it is produced from the PHGDH shunt. The increase in serine glycolytic synthesis is linked to the amplified activation of toll-like receptor 4 (TLR-4) after activation by lipopolysaccharide (LPS) [69] or hydrogen peroxide [75]. TLR-4 then induces a cytokine response, namely interleukin-1β [69]. Furthermore, GSH, whose activity and expression is upregulated by stress and higher levels of serine, further contributes to cytokine production and maintenance of redox balance in the cellular environment [69].

Serine deprivation was shown to be effective in reducing the activation of TLR-4, and reducing cytokine levels [69]. Activation of TLR-4 has also been implicated in the etiology of AMD [76]. Contrarily, serine supplementation is being explored as a potential therapy. Exogenous supplementation has proven to be effective in many cases reducing oxidative stress and reducing cytokine levels, but endogenous synthesis (as discussed above) augments the inflammatory response [77]. Therefore, balancing the anti-oxidant effects of serine with the pro-inflammatory nature of its endogenous synthesis is something to consider when developing a serine-based therapy for retinopathies in which inflammation contributes to the pathogenesis.

## 5. Consequences of Aberrations in Serine Metabolism

Our exploration has thus far presented the extensive involvement of serine metabolism in retinal homeostasis. Therefore, aberrations to this important keg in the metabolic machinery can negatively influence retinal pathologies. Glycolytic serine synthesis provides the largest contribution to intracellular serine stores, so deficiencies generally result from abnormalities in that synthetic process. For instance, deleterious mutations in PHGDH have been associated with microcephaly, reduced cognition, and psychomotor abnormalities [78]. Neu-Laxova is a fatal congenital disease marked by serious systemic abnormalities and is attributed to homozygous mutations in enzymes involved in serine biosynthesis [1]. This is recapitulated in the PHGDH^−/−^ mouse model, where mice suffer from embryonic lethality [1]. These irregularities can affect every part of the cell, from mitochondrial biosynthesis to oxidative imbalance.

An extensive comparative metabolomics study was performed by Gao et al., analyzing the metabolic and mitochondrial changes of colon cancer cell lines that occur as a result of serine deprivation [79]. Primarily, glycine, serine, threonine, pyrimidine, and sphingolipid pathways were the most significantly affected. Cells exhibited reduced fatty acid synthesis, and reduced TCA intermediates, consequently cells had a 45% reduction in ATP. Serine deprivation also caused major changes in mitochondrial membrane potential and increased fragmentation. However, phospholipid and phosphatidylserine levels remained comparably similar to standard culture conditions; sphingosine and ceramide levels on the other hand were significantly lowered [79]. It was concluded that mitochondrial fragmentation was exacerbated by reduced production of ceramides that serve as major constituents of the mitochondrial membrane. Ultimately, Gao et al. was able to isolate effected pathways that attenuated proper mitochondrial functions critical for cellular viability [79]. Considering the metabolic demands in the retina, such deficits would be extremely detrimental and could play a significant role in the etiology of many retinal diseases associated with metabolic dysregulation.

Cytotoxic aggregation of deoxysphingolipids is another important feature of serine deficiency [78]. As plasma serine levels decline, alanine flux increases and promotes the production of deoxysphingolipids [78]. As previously indicated, serine sphingolipids are synthesized through the condensation of serine and palmitoyl-CoA mediated by SPT. In cases of serine deficiency, SPT may incorporate alanine or glycine into the formation of the lipid, forming a cytotoxic analogue [78]. These lipids have been shown to aggregate and induce apoptosis in in vitro and in vivo models [78]. Retinal organoids treated with deoxysphingolipids exhibited dose-dependent apoptosis [9].

## 6. Retinal Degeneration and Dysregulated Serine Metabolism

### 6.1. Inherited Retinal Degeneration

Inherited retinal degeneration (IRD) can arise from mutations in various genes linked to photoreceptor development and structure or genes involved in phototransduction. To add to the complexity of IRDs, patients with similar mutations may present with very different phenotypes [80]. This is indicative of other underlying factors that can contribute to disease manifestation. It has been shown that metabolic dysregulation is a common element of various IRD models [81]. One example that has recently emerged is the dysregulation of sphingolipid metabolism [3]. Many investigators have shown that ceramide toxicity is elevated in various models of IRD [22]. In contrast, levels of the protective sphingolipid, sphingosine-1-phosphate, are significantly reduced [22]. What is yet to be determined is why and how the toxic ceramides are increased while sphingosine-1-phosphate levels diminish. One mechanism that has been postulated suggests that serine deficiency forces the sphingolipid metabolism to switch to the incorporation of alanine for sphingolipid synthesis [22]. However, in doing so, the reaction is skewed towards the formation of ceramides. Further investigation is required to determine why there is reduced serine availability. Since the glycolytic precursor, phosphoglycerate, is the primary source for serine biosynthesis, it is possible that dysregulation in glucose metabolism is the cause for the reduced serine availability.

### 6.2. Diabetic Retinopathy

Studies have demonstrated a correlation between serine deficiency and systemic diabetes [77]. Inflammation potentially increases the expression and activity of SRR, and consequently increases the availability of D-serine for receptor binding [40]. The elevated levels of D-serine contribute to glutamate toxicity and induce RGC apoptosis [40]. More precisely, as more D-serine binds to NMDARs, an excitotoxic response is induced [40]. As part of glutamate signal propagation in the retina, NMDAR excitation may be involved in decreasing neurotransmitter sensitivity [40]. Nevertheless, when D-serine levels are substantially elevated, the NMDAR-mediated excitation has a toxic effect in the retinal environment and induces cell death in RGCs [40]. In SRR null models with streptozocin (STZ) induced DR, degeneration of RGCs appeared attenuated, there was a reduction in neovascularization and the retina appeared significantly healthier than the control [40]. Most interestingly, multiple studies have found that if diabetic patients are treated with a serine supplement, blood glucose levels were reduced [77,82]. Currently, no mechanism is known to explain how this happens. But it is known that upon onset of diabetes, similar to serine deficiency, toxic sphingolipid accumulation starts occurring [83]. In fact, deoxysphingolipids have been shown to be potent biomarkers for diabetes mellitus [84]. As in IRD, DR results in similar toxic build-up of these aberrant sphingolipids, but this occurs far before any retinopathic complications start to develop [25]. These ceramides can actually be used as an early indicator of disease. It appears as if serine deprivation and toxic ceramide accumulation occur concomitantly. Perhaps, as the retina relies most heavily on serine, any deficiencies may cause the tissue to exhibit the first signs of disease in DR.

## 7. Novel Insights: Macular Telangiectasia

It has been elucidated recently that a dysregulated lipid metabolism can lead to increased and leaky vasculature [16]. Given the fact that serine homeostasis closely regulates lipid metabolism, it becomes a prime candidate for such monitoring. Interestingly, it was found in patients suffering from Mac Tel that their PHGDH activity is also compromised [85]. This led to severe serine deficiency, which contributed to vascular abnormalities [9]. As explained earlier, serine deficiency further leads to toxic deoxysphingolipid accumulation, which indeed plays a role in the etiology of Mac Tel type 2 [86]. However, administering exogenous serine to patients as a promising therapeutic approach has been marginally successful [87,88]. Importantly, how serine deficiency may affect the RPE and whether that has a role in Mac Tel is not yet known. Nevertheless, it is to be determined whether the restoration of physiological levels of serine in the retina can prevent further neovascularization in Mac Tel patients.

## 8. Concluding Remarks

Serine metabolism has vast interconnectivity with many of the homeostatic mechanisms that work in concert to maintain retinal health and function, as depicted in Figure 3. As reviewed above, considerable evidence has indicated that both the RPE and Müller cells of the inner retina have the requisite enzymes for serine biosynthesis. Besides the significant contribution of these two tissues to the serine pool of the entire retina, we also established the role de novo serine synthesis plays in redox currency generation and in mitigating free radical stress for both the neural retina and RPE. Given the retina is a metabolically high functioning tissue with high-energy demands and recent advancements indicating extra-mitochondrial contribution to the elevated presence of free radicals compared to other tissues, it calls for such extensive measures. Thus we show that while the pentose phosphate pathway might be sufficient to maintain the redox balance for other tissues, the retina depends upon additional tools, which it obtains primarily from serine metabolism: like NADPH from de novo biosynthesis, glycine, GSH and NADH. In addition, a review of the literature on serine-based lipid derivatives like phosphatidylserine, sphingolipids, and their toxic form, i.e., ceramides, helped us conclude the essential role these play in both RPE phagocytosis and membrane integrity in retinal homeostasis, while their imbalance is a critical factor in inherited retinal dystrophies. Coupled with the above observations and the recent advance on using D-serine as therapeutic candidate for diabetic retinopathy, we further postulate that there are a multitude of potential therapies targeting serine metabolism that hold tremendous promise against retinal diseases. We highlighted the prospect of using PHGDH replacement gene therapy or serine supplement therapy for Mac Tel patients and serine racemase and ceramide synthase inhibition for DR patients. Careful evaluation of recent literature also helped us align with the growing consensus that metabolic vulnerabilities add an extra layer of susceptibility for IRD patients. Thus, boosting serine biosynthesis and serine metabolism by pharmacological activators or complimentary gene therapy in these patients may reduce this risk factor and help postpone their onset of degeneration. Consequently, and in this review, our goal was to delineate some of the important roles of serine and demonstrate how they are imperative to the health of the retina. By providing a comprehensive view of the relationship between serine and retinal health, we hope to bring more awareness to the importance of serine to the retina so it can be further assessed for treatment options, and proteins that mediate its metabolic processes can be considered as viable targets for gene therapy. 

## Figures and Tables

**Figure 1 cells-09-00674-f001:**
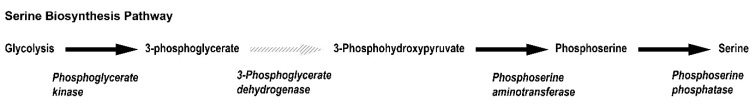
Pathway depicting serine biosynthesis from glycolysis. Metabolic intermediates involved in the enzymatic synthesis of L-serine from glycolysis is shown here with the rate limiting step marked with a dashed grey arrow. The enzymes involved in respective steps are shown in bold italics below the arrow for the individual reaction.

**Figure 2 cells-09-00674-f002:**
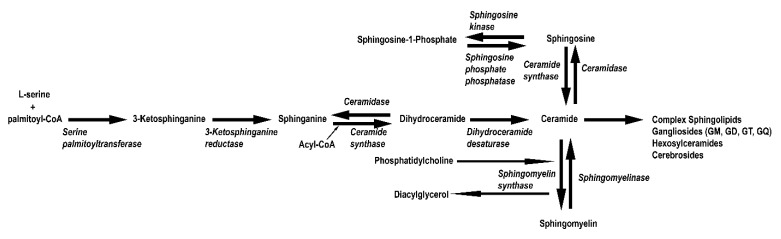
Pathway depicting sphingolipid biosynthesis from L-serine. Metabolic intermediates involved in the enzymatic synthesis of sphingolipids from L-serine are shown here. The enzymes involved in respective steps are shown in bold italics below the arrow for the individual reaction.

**Figure 3 cells-09-00674-f003:**
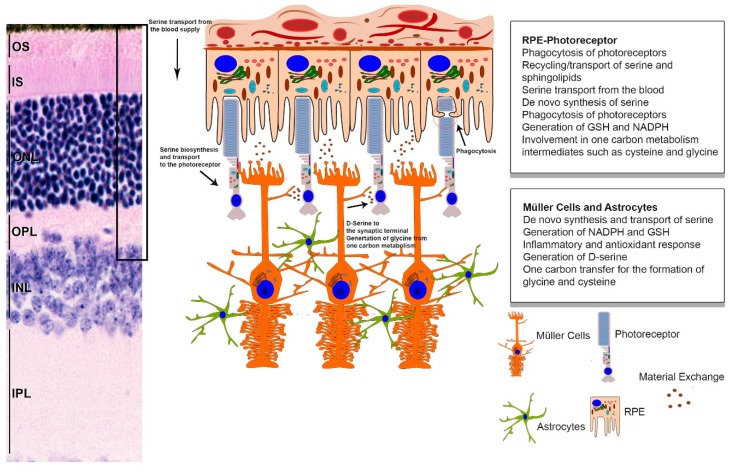
Graphical summary of serine metabolism in the retina. Serine homeostasis is primarily maintained by the RPE and retinal glia. The RPE transports and generates serine, which is ultimately transported to the photoreceptors. Additionally, important serine metabolic products such as glycine and cysteine are transported or catabolized to be used as fuel, as the energetic requirements of the RPE are very high. Photoreceptors also receive serine from the Müller glia and astrocytes. Glial cells are vital to the macula and generate serine from glycolysis, which is crucial in maintaining the redox balance in the photoreceptors, controlling neurotransmission, and mediating inflammation response elements. (IS, inner segment; OS, outer segment; ONL, outer nuclear layer; OPL, outer plexiform layer; INL, inner nuclear layer; IPL, inner plexiform layer).

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
