# Peer review of "The Intersection of Serine Metabolism and Cellular Dysfunction in Retinal Degeneration"

_cells, 2020, doi:10.3390/cells9030674_

Round 1

Reviewer 1 Report

This is an informative and well written minireview about serine in the biology of the neural retina and its supporting structures. It is approachable, well referenced, and appears to be a good contribution to the literature.

Minor suggestions for improvement are included:

Introduction, 3rd paragraph—reference 19 is listed for supporting a role for decreased serine levels in AMD; it’s a PRER review, is there a primary reference for this?

same paragraph: These sentences need a little light editing-

The retina is a complex stratified tissue consisting of the retinal pigment epithelium (RPE), a critical layer of cells for retinal homeostasis, and the neural retina contacting[SIC] the two types of photoreceptor cells. The neural retina also consist[SIC] of second order neurons and the retinal ganglion cell (RGC) layer that forms the optic nerve.

In the paragraph on Serine and RPE phagocytosis, I did not understand this statement:

Furthermore, the connections between phagocytosis and serine metabolism has been shown to be effective [30].

In the section on Serine and epigenetic regulation

The authors state “In AMD, changes to methylation of “risk alleles” in both the RPE and the neural retina have been attributed to disease pathogenesis [49]. ”

There is an earlier work by Hunter et al.on epigenetics and gene expression in AMD eyes (PMID 22410570) that should be cited here.

Section II discussion of macular parameters-it might be helpful for the authors to define a bit how they are using the term macula, since the anatomic macula and the clinical macula mean different things.

Figure 3- the inner and outer segment labels on the retina micrograph are flipped (OS should be next to the RPE).

The authors might also consider using a color photograph (toluidine blue thick section?) since the cartoon is already in color (so the authors are committed to a color plate) and that might be more aesthetically pleasing. 

Author Response

Reviewer#1

Minor suggestions for improvement are included:

  1. Introduction, 3rd paragraph—reference 19 is listed for supporting a role for decreased serine levels in AMD; it’s a PRER review, is there a primary reference for this?

Response: A references was added.

  1. same paragraph: These sentences need a little light editing-

The retina is a complex stratified tissue consisting of the retinal pigment epithelium (RPE), a critical layer of cells for retinal homeostasis, and the neural retina contacting[SIC] the two types of photoreceptor cells. The neural retina also consist[SIC] of second order neurons and the retinal ganglion cell (RGC) layer that forms the optic nerve.

Response: Manuscript edited as suggested.

  1. In the paragraph on Serine and RPE phagocytosis, I did not understand this statement:

Furthermore, the connections between phagocytosis and serine metabolism has been shown to be effective [30].

Response: Statement has been edited.

  1. In the section on Serine and epigenetic regulation

The authors state “In AMD, changes to methylation of “risk alleles” in both the RPE and the neural retina have been attributed to disease pathogenesis [49]. ” There is an earlier work by Hunter et al.on epigenetics and gene expression in AMD eyes (PMID 22410570) that should be cited here.

Response: Article is cited as suggested.

  1. Section II discussion of macular parameters-it might be helpful for the authors to define a bit how they are using the term macula, since the anatomic macula and the clinical macula mean different things.

Response: The term macula has now been clarified as suggested.

  1. Figure 3- the inner and outer segment labels on the retina micrograph are flipped (OS should be next to the RPE).

The authors might also consider using a color photograph (toluidine blue thick section?) since the cartoon is already in color (so the authors are committed to a color plate) and that might be more aesthetically pleasing.

Response: We apologize for the oversight. A color figure is used in the revised version.

Reviewer 2 Report

This manuscript provides rather general information on the serine metabolism in general and attempts to relate it to retinal pathologies, but these attempts are unsuccessful as not a clear conclusion on the subject of this manuscript can be drawn. This manuscript is not logically coherent as many aspects are repeated and some facts are not in their right places in the text. It provides rather general information, mostly textbook-like.

Key words: Mac Tel is rather too specific term to be placed in keywords, better full name

Abstract: Serine is a non-essential amino acid occurring in the most, if not all, proteins operating in the retina. Therefore, the authors should specify, why this amino acid is particularly important in retina functioning, e.g. they should specify what serine has to do with phagocytosis of POS. In other words they should stress that serine is not only an essential component of many retina-important proteins, but also plays a significant role as an amino acid bound to other objects or mediator of vital retina-related processes in its free form.

Introduction

It is not clear whether the authors intend to consider the entire retina or limit their considerations to the neural retina.

The section I. Why is serine important to the retina? does not convince me that it is particularly important, i.e. it is of superior over other amino acids.

The paragraph on general DNA methylation and its specific significance in DR and AMD (page 8) is imprecise. What has cancer to do with the retina? This is trying to explain in the next section, but the statement: “many of the observation addressed in this section have been obtained from cancer cell lines.” is at least surprising.

The section “V. Consequences of aberrations in serine metabolism” is somehow non-consequent, as the authors tied to show that in the previous sections.

The subsequent section: “IV. Retinal degeneration and dysregulated serine metabolism” is limited to Inherited retinal degeneration and Diabetic retinopathy, although the authors frequently related to AMD in the preceding sections.

Concluding remarks – this section is practically empty and does not contribute anything new for that manuscript.

Author Response

Reviewer#2

  1. Key words: Mac Tel is rather too specific term to be placed in keywords, better full name

Response: This has been edited as suggested.

  1. Abstract: Serine is a non-essential amino acid occurring in the most, if not all, proteins operating in the retina. Therefore, the authors should specify, why this amino acid is particularly important in retina functioning, e.g. they should specify what serine has to do with phagocytosis of POS. In other words they should stress that serine is not only an essential component of many retina-important proteins, but also plays a significant role as an amino acid bound to other objects or mediator of vital retina-related processes in its free form.

Response: This has been edited as suggested

  1. Introduction

It is not clear whether the authors intend to consider the entire retina or limit their considerations to the neural retina.

Response: We have made sure to emphasize that the review is focused on the entire retina (i.e. neural retina-RPE) and only those sections which are focused on the neural retina have the labels as “neural retina”.

  1. The section I. Why is serine important to the retina? does not convince me that it is particularly important, i.e. it is of superior over other amino acids.

Response: The reviewer is correct that other amino acids may also be important. However, we have not indicated that the other amino acids are not important. We are reviewing the literature about the role of serine.

  1. The paragraph on general DNA methylation and its specific significance in DR and AMD (page 8) is imprecise. What has cancer to do with the retina? This is trying to explain in the next section, but the statement: “many of the observation addressed in this section have been obtained from cancer cell lines.” is at least surprising.

Response: This is rather a surprising statement. Research on cancer and cancer cell lines has provided most of the knowledge in molecular/cell biology and biochemistry. We are attracting attention to an area of research on the role of serine that has not been well investigated in the retina.

  1. The section “V. Consequences of aberrations in serine metabolism” is somehow non-consequent, as the authors tied to show that in the previous sections.

Response. In light of the fact the other two reviewers did not have an issue with this section, we maintained the section as is.

  1. The subsequent section: “IV. Retinal degeneration and dysregulated serine metabolism” is limited to Inherited retinal degeneration and Diabetic retinopathy, although the authors frequently related to AMD in the preceding sections.

Response: It is not clear what the issue is.

  1. Concluding remarks – this section is practically empty and does not contribute anything new for that manuscript.

Response. In light of the fact the other two reviewers do not agree with this reviewer’s comment, we maintained the section as is.

Reviewer 3 Report

Dear Editor,

The present study from Sinha et al.,   is of undoubted interest. Important aspects of this topic that appears novel have been touched, and these appear quite well conducted.  Literature is sufficiently up-to date. 

I ams positive about the fact that glial cells provide many metabolites to support surrounding neuron cells including serine, and that GSH metabolism is vital especially in neurons (for having unpublished MS and WGCNA data on this topic) and that Glycine supply is consequently very important, being GSH a  major downstream product of de novo serine synthesis metabolism in central nervous system. However, the manuscript is overall too focused exclusively on serine as the first and last resource for the retina.

However the Authors fail to show why photoreceptors should need this supply more than other parts of the retina, if not for the well-known notion that retina is the most demanding tissue in the body and in particular photoreceptors, one of the most metabolically active cell types in the body (Poitry-Yamate et al., 1995). On the other hand, oxygen is absorbed mainly at the level of the photoreceptors (Invest Ophthalmol Vis Sci 31:1029-1034,1990). Clearly Reactive Oxygen Species (ROS) must be generated in the sites that consume oxygen the most.

ROS are generated from either mitochondrial (mainly the respiratory chains) or nonmitochondrial sources, including NADPH, xanthine oxidase and others, but Authors appear to ignore that a major extra-mitochondrial ROS source is the ectopic oxidative phosphorylation that occurs in the rod Outer Segments and in myelin . The ectopic respiratory chain would be a primary unshielded and unsuspected source of Reactive Oxygen Species. Recent evidence shows that the primary retinal damage from oxidative stress production occurs to rod outer segments and not only in the inner segment containing the mitochondria, but also in the Outer limb (Funk, Roelke et al.). Consistently, glycine and thus serine supply play a pivotal antioxidant role in the rods.

Taking into consideration these emerging data on a source of oxidative stress inside the photoreceptor outer segments, besides mitochondria, the work could provide an advance towards the current knowledge.

My best regards

Author Response

Reviewer#3

  1. The present study from Sinha et al.,   is of undoubted interest. Important aspects of this topic that appears novel have been touched, and these appear quite well conducted. Literature is sufficiently up-to date. I am positive about the fact that glial cells provide many metabolites to support surrounding neuron cells including serine, and that GSH metabolism is vital especially in neurons (for having unpublished MS and WGCNA data on this topic) and that Glycine supply is consequently very important, being GSH a major downstream product of de novo serine synthesis metabolism in central nervous system. However, the manuscript is overall too focused exclusively on serine as the first and last resource for the retina.

Response: We appreciate the reviewer’s comment. In writing this review we are intending to focus the attention on the role serine plays in retina. However, we have not played down the role of any other amino acids or metabolite.

  1. However the Authors fail to show why photoreceptors should need this supply more than other parts of the retina, if not for the well-known notion that retina is the most demanding tissue in the body and in particular photoreceptors, one of the most metabolically active cell types in the body (Poitry-Yamate et al., 1995). On the other hand, oxygen is absorbed mainly at the level of the photoreceptors (Invest Ophthalmol Vis Sci 31:1029-1034,1990). Clearly Reactive Oxygen Species (ROS) must be generated in the sites that consume oxygen the most. ROS are generated from either mitochondrial (mainly the respiratory chains) or nonmitochondrial sources, including NADPH, xanthine oxidase and others, but Authors appear to ignore that a major extra-mitochondrial ROS source is the ectopic oxidative phosphorylation that occurs in the rod Outer Segments and in myelin . The ectopic respiratory chain would be a primary unshielded and unsuspected source of Reactive Oxygen Species. Recent evidence shows that the primary retinal damage from oxidative stress production occurs to rod outer segments and not only in the inner segment containing the mitochondria, but also in the Outer limb (Funk, Roelke et al.). Consistently, glycine and thus serine supply play a pivotal antioxidant role in the rods. Taking into consideration these emerging data on a source of oxidative stress inside the photoreceptor outer segments, besides mitochondria, the work could provide an advance towards the current knowledge.

Response: We appreciate the reviewer’s detailed comments and have added the suggested references as well as include details on extra-mitochondrial ROS and importance of serine/glycine for the same.

Round 2

Reviewer 2 Report

The authors did not address several my remarks stating that the other reviewers had not objections to the fragments I questioned.

However, the revised manuscript still requires changes in several fragments, e.g. although the "Serine and epigenetic regulation" section is not so strongly related to cancer, it presents textbook general information and it is not presented how, or even whether, serine is involved in specific regulation in retina diseases. The two last statement about DR and AMD has nothing to do with disease-specific changes in serine metabolism. Several other impreciseness should be corrected throughout manuscript, e.g. in the same section: "Epigenetic methylation..." - should be rather DNA methylation as the authors meant that and not histone methylation. Moreover, DNA methylation does not always suppresses gene expression that depends on whole epigenetic pattern.

I insist for rewriting the Concluding remarks section as it does not contain any conclusion related to specific subjects considered in the previous sections, e.g. epigenetics, redox balance, potential therapeutic target in the retina diseases (is the same for all diseases?), inflammation vs. antioxidant...

Author Response

March 5, 2020

Ms. Christina Zhang

Editor, MDPI

RE: Cells-725010. The intersection of serine metabolism and cellular dysfunction in retinal degeneration

Dear Ms. Zhang:

Thank you for providing us with the re-reviewer comments of our above referenced article. Please find below the response to the reviewer’s comments.

We trust that the manuscript, in its re-revised form, is acceptable for publication.

Sincerely,

Muayyad R. Al-Ubaidi, Ph.D.

Reviewer#2

The authors did not address several my remarks stating that the other reviewers had not objections to the fragments I questioned.

However, the revised manuscript still requires changes in several fragments, e.g. although the "Serine and epigenetic regulation" section is not so strongly related to cancer, it presents textbook general information and it is not presented how, or even whether, serine is involved in specific regulation in retina diseases. The two last statement about DR and AMD has nothing to do with disease-specific changes in serine metabolism. Several other impreciseness should be corrected throughout manuscript, e.g. in the same section: "Epigenetic methylation..." - should be rather DNA methylation as the authors meant that and not histone methylation. Moreover, DNA methylation does not always suppresses gene expression that depends on whole epigenetic pattern.

I insist for rewriting the Concluding remarks section as it does not contain any conclusion related to specific subjects considered in the previous sections, e.g. epigenetics, redox balance, potential therapeutic target in the retina diseases (is the same for all diseases?), inflammation vs. antioxidant...

Response: to address the reviewer’s comments we have:

Rephrased the paragraph regarding how/if serine is involved in specific regulation in retinal diseases as below:

The link between epigenetic modulations and retinal disease is currently at a nascent phase, so correlations between aberrant serine metabolism, DNA-methylation, and retinal disease remain unclear. However, recent studies performed on 3 pairs of monozygotic twins with different presentations of AMD indicated significant changes to DNA methylation patterns of genes that may be implicated in disease pathogenesis. More critically, the diet of the studied twins also highlighted nutritional significance in epigenetic regulation. Subjects with reduced dietary methionine, vitamin D, and betaine had worse disease prognosis, implicating the importance of nutrient bioavailability and epigenetics. As serine is involved in the synthesis of methionine, further studies should explore serine levels in retinal disease in relation with epigenetic changes that may contribute to disease onset and/or progression.

Deleted the sentence about DR and AMD.

Corrected the text, re: “"Epigenetic methylation..." - to be more precise, it isnow “DNA methylation”

Modified the conclusion as below:

Serine metabolism has vast interconnectivity with many of the homeostatic mechanisms that work in concert to maintain retinal health and function, as depicted in Figure 3. As reviewed above, considerable evidence has indicated that both the RPE and Müller cells of the inner retina have the requisite enzymes for serine biosynthesis. Besides the significant contribution of these two tissues to the serine pool of the entire retina, we also established the role de novo serine synthesis plays in redox currency generation and in mitigating free radical stress for both neural retina and RPE. Given the retina is a metabolically high functioning tissue with high-energy demands and recent advancements indicating extra-mitochondrial contribution to the elevated presence of free radicals compared to other tissues, it calls for such extensive measures. Thus we show that while pentose phosphate pathway might be sufficient to maintain the redox balance for other tissues, the retina depends upon additional tools which it obtains primarily from serine metabolism: like NADPH from de novo biosynthesis, glycine, GSH and NADH. Also reviewed the literature on serine based lipid derivatives like phosphatidylserine, sphingolipids, and their toxic form i.e. ceramides, helped us conclude the essential role these play in both RPE phagocytosis and membrane integrity in retinal homeostasis while their imbalance being a critical factor in inherited retinal dystrophies. Coupled with the above observations and the recent advance on using D-serine as therapeutic candidate for diabetic retinopathy, we further postulate that there are a multitude of potential therapies targeting serine metabolism that hold tremendous promise against retinal diseases. We highlighted the prospect of using PHGDH replacement gene therapy or serine supplement therapy for Mac Tel patients and serine racemase and ceramide synthase inhibition for DR patients. Careful evaluation of recent literature also helped us align with the growing consensus that metabolic vulnerabilities add an extra layer of susceptibility for IRD patients. Thus boosting serine biosynthesis and serine metabolism by pharmacological activators or complimentary gene therapy in these patients may reduce this risk factor and help postpone their onset of degeneration. Consequently and in this review, our goal was to delineate some of the important roles of serine and demonstrate how they are imperative to the health of the retina. By providing a comprehensive view of the relationship between serine and retinal health, we hope to bring more awareness to the importance of serine to the retina so it can be further assessed for treatment options, and proteins that mediate its metabolic processes can be considered as viable targets for gene therapy.